# Effect of the Wetting Hydraulic Property of Soil on 1-D Water Infiltration

Xuebo Li [1], Tianlun Shen [2], Ke Xiang [2], Qian Zhai [2,*], Harianto Rahardjo [3], Alfrendo Satyanaga [4] and Shijun Wang [2]

1   School of Civil Engineering, Tianjin University, Tianjin 300072, China
2   Key Laboratory of Concrete and Prestressed Concrete Structures of Ministry of Education, Bridge Engineering Research Center of Southeast University, Southeast University, Nanjing 210096, China
3   School of Civil and Environmental Engineering, Nanyang Technological University, Block N1, Nanyang Ave., Singapore 639798, Singapore
4   Department of Civil and Environmental Engineering, School of Engineering and Digital Sciences, Nazarbayev University, Kabanbay Batyr Ave., 53, Nur-Sultan 010000, Kazakhstan
*   Correspondence: zhaiqian@seu.edu.cn

**Abstract:** Rainwater infiltration is primarily governed by the soil-water characteristic curve (SWCC) and hydraulic conductivity function (HCF) of soil. Both the SWCC and the HCF are hysteretic during the drying and wetting processes. In a numerical simulation, different seepage results can be obtained by incorporating different hydraulic conductivity functions of soil. In practice, the wetting HCF is commonly estimated from the wetting SWCC using the statistical method, which is named $HCF_{swcc,w}$ in this note. However, there is no study that has verified the results from seepage analyses using $HCF_{swcc,w}$. Therefore, the objective of this study is to investigate the influence of wetting SWCC and wetting HCF on 1-D water infiltration. The results from the numerical simulations were verified with the instrumentation reading from a soil column. It was observed that the results from the model using wetting $HCF_{PSDF}$, which defines the wetting HCF estimated using the concept of pore-size distribution function, gave better agreement with the instrumented data. Therefore, both wetting SWCC and wetting $HCF_{PSDF}$ are advised to be used as input information for the numerical simulation of rainwater infiltration.

**Keywords:** simulation of rainwater infiltration; wetting SWCC for infiltration analysis; wetting HCF for infiltration analysis; hysteresis of SWCC; infiltration to soil column

## 1. Introduction

Hydraulic conductivity function (HCF), which defines the relationship between hydraulic conductivity and soil suction, is commonly used as the input information for seepage analysis [1]. Different researchers have proposed various models for the estimation of the HCF from a soil-water characteristic curve (SWCC). Leong and Rahardjo [2] indicated that, among those models, the theorical background of the statistical method was most rigorous. As a result, the statistical method has been widely used for the estimation of the HCF from a SWCC by different researchers (Childs and Collis-George [3], Mualem [4], Fredlund et al. [5,6], Zhai and Rahardjo [7], and Zhai et al. [8]). Popa et al. [9], Zelenakova et al. [10,11], and Elewa et al. [12] have studied the spatial distribution of groundwater and the surface runoff in the rural area. A correct SWCC and HCF are crucial for the evaluation of the groundwater table and surface runoff.

It is known that the water content and hydraulic conductivity (HC) corresponding to the suction in the drying process are higher than those in the wetting process; this phenomenon is commonly referred to as "hysteresis". The measurement of wetting HCF is more difficult than the measurement of wetting SWCC. Therefore, the wetting HCF is commonly estimated rather than directly measured. As the statistical method has been

widely used for the estimation of the drying HCF from the drying SWCC, this method is also adopted to estimate the wetting HCF from the wetting SWCC (HCF$_{swcc,w}$). However, the accuracy of this approach has not been verified in past studies. In this note, the limitations of the statistical method in the estimation of the wetting HCF from the wetting SWCC are explained. Subsequently, both HCF$_{swcc,w}$ and wetting HCF$_{PSDF}$, which define the estimated wetting HCF based on the concept of the pore-size distribution function (PSDF), were adopted in the numerical simulation for the one-dimensional (1D) infiltration into the soil column. The results from the models with both HCF$_{swcc,w}$ and wetting HCF$_{PSDF}$ are compared with the measurement data from a soil column test.

## 2. Materials and Methods

Childs and Collis-George [3] were the first to propose the statistical method for the estimation of HCF from SWCC. Zhai et al. [13] indicated that there were four major assumptions adopted in the statistical method: (i) the SWCC is analogous to the pore-size distribution function (PSDF); (ii) the pores in the soil are simplified as a series of capillary tubes with different sizes; (iii) the capillary tubes are randomly distributed in soil; and (iv) the capillary tubes are randomly connected with each other. Zhai and Rahardjo [7] proposed a general equation for the statistical method, as shown in Equation (1), for the estimation of the HCF from SWCC. Substituting the SWCC equation, the HCF can be calculated directly from the fitting parameters of those SWCC equations.

$$k(\psi_x) = k(\psi_{ref}) \frac{\left\{ \sum_{i=x}^{N} \left[ \frac{(S(\psi_x)-S(\psi_i))^2 - (S(\psi_x)-S(\psi_{i+1}))^2}{(\psi_i)^2} \right] \right\}}{\left\{ \sum_{i=ref}^{N} \left[ \frac{\left(S(\psi_{ref})-S(\psi_i)\right)^2 - \left(S(\psi_{ref})-S(\psi_{i+1})\right)^2}{(\psi_i)^2} \right] \right\}} \tag{1}$$

where $k(\psi_x)$ = calculated the HC with given suction of $\psi_x$; $k(\psi_{ref})$ = HC at the reference point(i.e., $\psi = \psi_{ref}$); $\psi_{ref}$ = the suction corresponding to the reference point; $S(\psi_{ref})$ = degree of saturation corresponding to the reference point, $\psi_x$; $\psi_i$ = the soil suction in the drying process, $S(\psi_x)$; $S(\psi_i)$= degree of saturation corresponding to the soil suctions of $\psi_x$ and $\psi_i$, respectively; and $N$ = the total number of the divided SWCC segments.

Zhai and Rahardjo and Zhai et al. [7,13] indicated that the first assumption was most crucial for the application of the statistical method. Fredlund and Xing [5,6] indicated that the SWCC could be integrated from the PSDF only when the soil did not undergo volume change. In addition, the probability of the connection between the pores with different sizes is computed from the pore-size density. The pore-size density can be obtained from PSDF, which can be estimated from the drying SWCC. As a result, the accuracy in the representation of the PSDF is crucial in the estimation of the HCF by using the statistical method. Zhai et al. [14] indicated that wetting SWCC could not represent PSDF and that drying SWCC could represent the PSDF. The differences between drying and wetting SWCCs are the results of the "ink-bottle" and "rain-drop" effects but are not due to different PSDFs. Therefore, the wetting HCF should not be estimated from the wetting SWCC using the statistical method.

## 3. Results and Discussions

Klute [15] and Zhai et al. [14] demonstrated the schematic diagram of SWCCs in both the drying and wetting processes. Due to the entrapped air, the soil cannot be fully saturated after the wetting process, resulting in an open loop as shown in Figure 1. If the soil is re-desaturated again after the wetting process, the main drying curve is obtained. If the soil is saturated or desaturated at any point on the main drying curve (or on the main wetting curve), then the scanning curve is obtained.

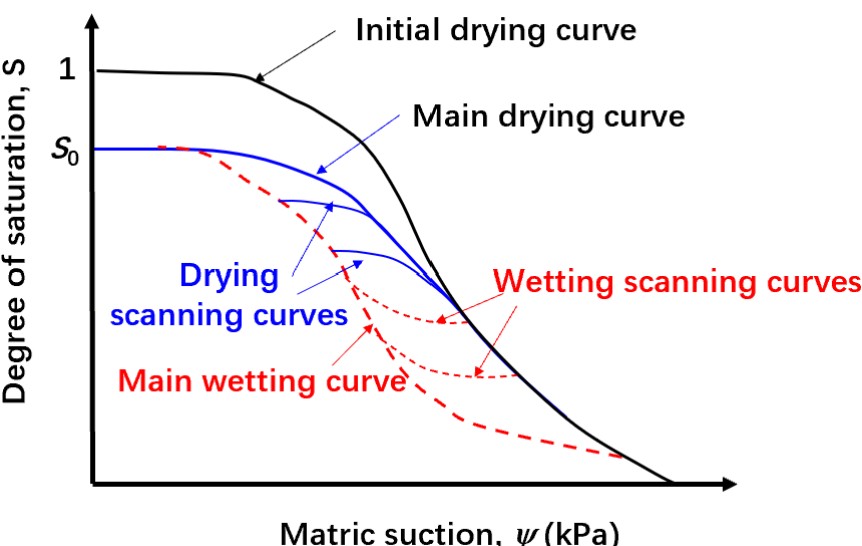

**Figure 1.** Schematic diagram of hysteresis in SWCC of a coarse-textured soil with rigid structure (modified from Klute [15] and Zhai et al. [14]).

It is known that when water drains out from soil, it does so progressively from the largest pores to the smallest pores. When the radius of the meniscus is larger than the pore radius, water cannot drain out of the pore and the pore remains wet. On the other hand, when the radius of the meniscus is smaller than the pore radius, water drains out the pore and the pore becomes dry. Therefore, the measured initial drying SWCC can reflect the pore-size distribution correctly in soil.

If the entrapped air is uniformly distributed from the largest pores to the smallest pores, then the main drying curve can be scaled from the initial drying curve (or the primary drying curve). It is noted that the main wetting curve has a different shape to the primary drying curve. Zhai et al. [14] indicated that the different shape in the wetting SWCC is mainly due to the entrapped air, the "rain-drop" effect, and the "ink-bottle" effect. As a result, the wetting SWCC is not analogous to the PSDF, which renders the major assumption adopted in the statistical method invalid for the wetting process. Therefore, it was concluded that the wetting HCF should not be estimated from the wetting SWCC using the statistical method.

Zhai et al. [16] adopted the starting point in the wetting process as the reference and proposed Equation (2) for the estimation of the wetting HCF by incorporating the "ink-bottle" and "rain-drop" effects.

$$
k(\psi_{x,w}) =
\begin{cases}
k(\psi_{ref,w}) & \text{when } \psi_{x,w} > \frac{\psi_{ref,w}}{k} \\
\text{Otherwise,} \\
k(\psi_{ref,w}) \dfrac{\left\{\left[\displaystyle\sum_{j=x,w}^{m-1}\dfrac{(S(k\psi_{x,w})-S(k\psi_j))^2-(S(k\psi_{x,w})-S(k\psi_{j+1}))^2}{(k\psi_j)^2}\right]S(\psi_j)\right\}+\left\{\displaystyle\sum_{i=ref,w}^{N}\dfrac{(S(\psi_{ref,w})-S(\psi_i))^2-(S(\psi_{ref,w})-S(\psi_{i+1}))^2}{(\psi_i)^2}\right\}}{\left\{\displaystyle\sum_{i=ref,w}^{N}\left[\dfrac{(S(\psi_{ref,w})-S(\psi_i))^2-(S(\psi_{ref,w})-S(\psi_{i+1}))^2}{(\psi_i)^2}\right]\right\}}
\end{cases}
\tag{2}
$$

where $k(\psi_{x,w})$ is the hydraulic conductivity with respect to the suction of $\psi_{x,w}$ in the wetting process; $k(\psi_{x,w})$ is the reference hydraulic conductivity (the hydraulic conductivity of the soil when it starts to be saturated); $k$ is the parameter in Zhai et al.'s model [14]; $\psi_{ref,w}$ is the suction (which is corresponding to $k(\psi_{ref,w})$) when the soil starts to be saturated; $\psi_{x,w}, \psi_i$ and

$\psi_j$ are the soil suctions; $S(k\psi_{x,w})$, $S(k\psi_j)$,$S(\psi_{ref,w})$, and $S(\psi_i)$ are the degrees of saturation on the drying SWCC with respect to the soil suctions of $k\psi_{x,w}$, $k\psi_j$, $\psi_{ref,w}$, and $\psi_i$, respectively; and *N* is the total number of SWCC segments.

As shown in Equation (2), in the wetting process the suction range of [*0*, $\psi_{ref,w}$] is divided into two zones such as [*0*, $\psi_{ref,w}/k$) and [$\psi_{ref,w}/k$, $\psi_{ref,w}$]. In the suction zone of [$\psi_{ref,w}/k$, $\psi_{ref,w}$], the water distribution in pores was not significantly changed. The contact angle changed from the receding contact angle to the advancing contact angle when the suction decreased from $\psi_{ref,w}$ to $\psi_{ref,w}/k$. When the suction decreased to a value less than $\psi_{ref,w}/k$, water started to fill the pores and the water distribution in the pores changed. However, because of the blocking of the larger dry pores, some of the small pores could not be filled, which is commonly referred to as "ink-bottle" phenomenon. $S(\psi_j)$ was applied in the computation of the wetting HCF as shown in Equation (2). The wetting HCF computed from Equation (2) was based on the concept of PSDF and that is why the wetting HCF estimated from Equation (2) is named as the wetting HCF$_{PSDF}$ in this note.

To evaluate the accuracy of the numerical model by using HCF$_{swcc,w}$ and the wetting HCFPSDF, numerical analyses for water infiltration into a soil column following the column set up from Ng et al. [17] were conducted. The measured suction profiles at different time stages can be used to verify the results from the numerical analyses. The column had an inner diameter of 140 mm and a height of 1 m. There were three layers of soil: 400 mm-thick Silt, 200 mm-thick Gravelly Sand, and 400 mm-thick Clay. These were stacked from the top to the bottom of the column, as shown in Figure 2. Both the measured drying and wetting SWCCs for the Silt are illustrated in Figure 3a. The measured saturated HC for Silt was $1.4 \times 10^{-6}$ m/s. Subsequently, both the drying SWCC and the saturated HC were adopted to estimate the drying HCF by using Equation (1); the obtained result was named as HCF$_{swcc,d}$, as illustrated in Figure 3b. Meanwhile, both the wetting SWCC and the saturated HC were adopted to estimate the drying HCF by using Equation (1); the obtained result was named as HCF$_{swcc,w}$, as illustrated in Figure 3b.

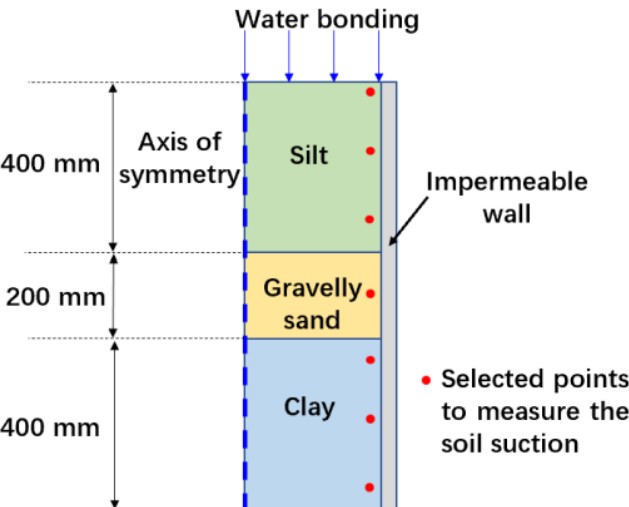

**Figure 2.** Illustration of the setting up of the infiltration into soil column from Ng et al. [12].

Zhai et al. [14] indicated that the shape of the wetting SWCC is a function of the initial suction when the soil starts to be saturated. In other words, different wetting SWCCs can be obtained if the soil was saturated at different suction levels. It was observed that initial suction in the Silt layer was around 45 kPa, which is different to the wetting SWCC as measured (initial suction of 500 kPa). To be consistent with the infiltration test, the scanning wetting SWCC with an initial suction of 45 kPa was estimated using the method from Zhai et al. [14]. Subsequently, the wetting HCF$_{PSDF}$ was estimated using Equation (2). Both the scanning wetting SWCC and the wetting HCF$_{PSDF}$ are illustrated in Figure 4a,b, respectively.

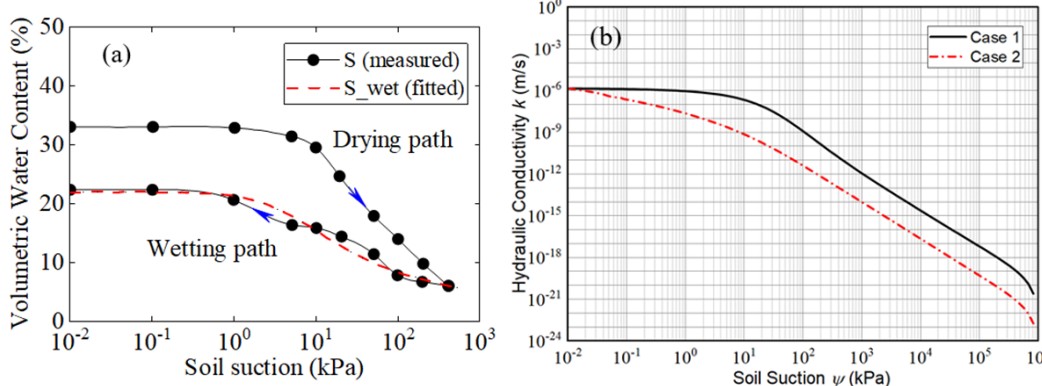

**Figure 3.** Illustration of the hydraulic properties of Silt and boundary conditions for the numerical simulation: (**a**) measured SWCCs (from Ng et al. [12]); (**b**) Estimated HCFs.

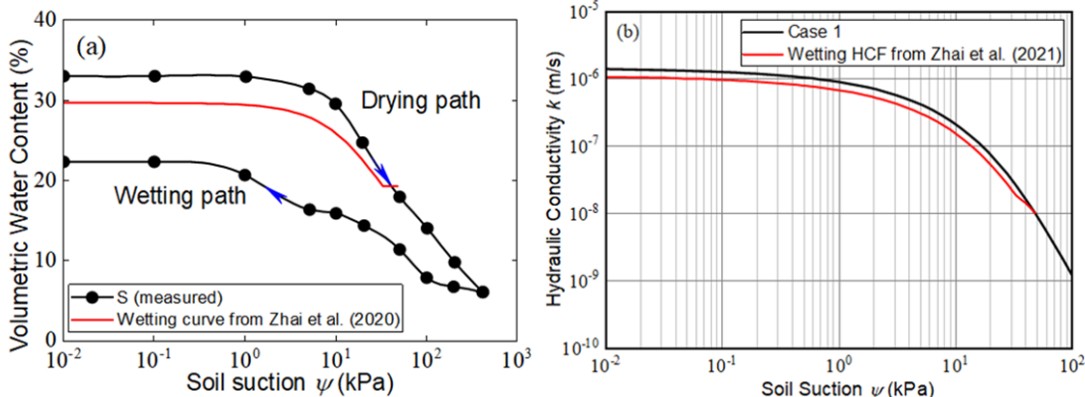

**Figure 4.** Illustration of the estimated scanning wetting SWCC and wetting HCF$_{PSDF}$: (**a**) Scanning wetting SWCC; (**b**) wetting HCF$_{PSDF}$. (from Zhai et al. (2021) [16]).

There was a total of three cases considered in the simulation: Case 1 refers to the numerical model incorporating both drying SWCC and HCF$_{swcc,d}$, ignoring hysteresis of the hydraulic properties of soil; Case 2 refers to the numerical model incorporating both wetting SWCC and HCF$_{swcc,w}$, which is the method adopted by most engineers including the hysteresis of hydraulic properties in the seepage analysis; Case 3 refers to the numerical model incorporating both scanning wetting SWCC and wetting HCF$_{PSDF}$, which considers the initial suction in the soil to be infiltrated. As water infiltration into the Silt layer is the main objective of this note, the hydraulic properties of Gravelly sand and Clay from the study by Ng et al. [17] were adopted for the seepage analyses, but they are not illustrated in this note. Both the measured and the computed suction profiles after 4 h and 6 h water infiltration were compared; these are illustrated in Figure 5.

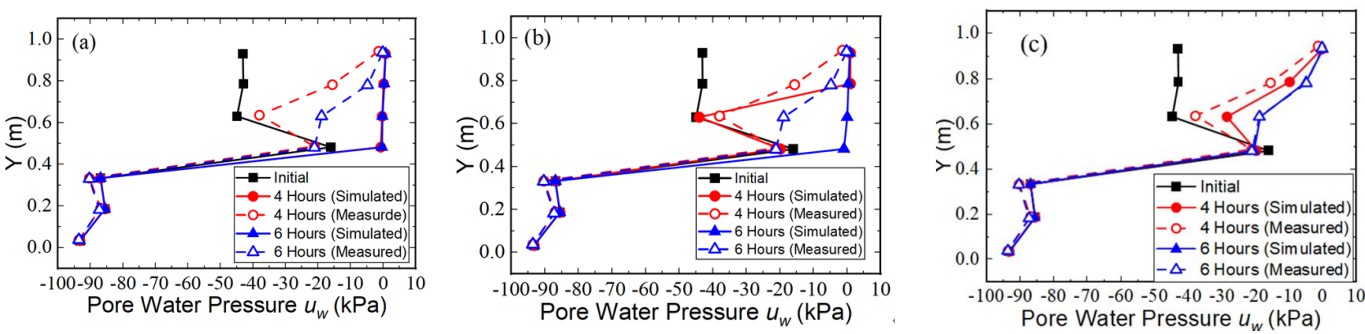

**Figure 5.** Comparisons between the measured and the simulated suction profiles after 4 h and 6 h water infiltration: (**a**) for Case 1; (**b**) for Case 2; (**c**) for Case 3.

Figure 5a indicates that the Silt in the numerical simulation was fully saturated after 4 h and 6 h water infiltration, while Silt in the 1-D infiltration column test (based on the instrumentation data) had a suction of around 20–30 kPa. It indicates that Case 1 (ignoring the hysteresis of the hydraulic properties) overestimates water infiltration. Therefore, it can be deduced that the hysteresis of the hydraulic properties should be incorporated in the seepage analysis for water infiltration. However, there are no past studies indicating a proper method to incorporate the hysteresis of the hydraulic properties in the seepage analysis. Figure 5b indicates that the suction profiles in Silt after 4 h of water infiltration as computed from Case 2 agree well with the measured data. However, the suction profiles in Silt after 6 h water infiltration as computed from Case 2 show obvious deviation from the measured data. This indicates that Case 2 is not a reliable model for the simulation of 1-D water infiltration. Figure 5c illustrates that the suction profiles in Silt after 4 h and 6 h water infiltration as computed from Case 3 agree well with the measured data. It seems that Case 3 provides the most reliable results of the column infiltration test.

Figure 5 indicates that measured wetting SWCC cannot be directly used for seepage analysis of water infiltration. The wetting scanning curve should be estimated from both the measured wetting SWCC and the initial suction in the soil to be infiltrated. The wetting $HCF_{PSDF}$ estimated on the concept of PSDF provided the best performance in the simulation of water infiltration compared with the performances of $HCF_{swcc,d}$ and $HCF_{swcc,w}$.

## 4. Conclusions

The correct hydraulic properties of soil are vital for a numerical simulation of water infiltration. The theoretical background of the statistical method for the estimation of HCF from both drying and wetting SWCCs was reviewed and discussed. There were limitations associated with estimating the wetting HCF from the wetting SWCC using the statistical method. Based on the analyses in this note, it was observed that measured wetting SWCC cannot be directly used for the seepage analysis of water infiltration. Different wetting scanning curves would be obtained when the soil is saturated with different initial suctions. Therefore, the initial suction in the soil to be infiltrated should be considered when estimating the wetting scanning curve. The model with both the wetting scanning curve and the wetting $HCF_{PSDF}$ gave the most reliable results on water infiltration.

**Author Contributions:** Methodology, H.R.; validation, A.S.; formal analysis, S.W.; investigation, K.X.; writing—original draft preparation, X.L.; writing—review and editing, T.S.; supervision, Q.Z. All authors have read and agreed to the published version of the manuscript.

**Funding:** This research received no external funding.

**Institutional Review Board Statement:** Not applicable.

**Informed Consent Statement:** Not applicable.

**Data Availability Statement:** Not applicable.

**Conflicts of Interest:** The authors declare no conflict of interest. The authors declare that they have no known competing financial interests or personal relationship that could have influenced the work reported in this paper.

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
