# Peer review of "Effect of the Wetting Hydraulic Property of Soil on 1-D Water Infiltration"

_sustainability, doi:10.3390/su15031822_

Round 1

Reviewer 1 Report

There are many failing of the manuscript from the previous revision that have not been addressed. Therefore, go back through those comments and addressed them properly. There are also further issues with the manuscript that are listed below. No progress has been made progress with this revision. English language and grammar require extensive revisions throughout the manuscript.

Lines 30 to 40 - You keep using the phrase 'different researchers'. This is not appropriate language and does not help set the right tone for a journal article.

Line 41 - You say 'it is known' but fail to provide a reference for that statement. You need to provide a reference.

There isn't any reference for lines 41 to 54. I doubt that this is findings from your current manuscript. If references are not provided for this section, it could be classed as plagiarism

Section 2 - There is too much background information in this section to constitute a methods section. If this information is required here, you need to write better. Currently the section reads more like an introduction.

 Line 90 - Again, you say 'it is known' but do not provide a reference. This could be classed as plagiarism is you are not careful. Also the term 'it is known' is not acceptable for a scientific journal.

Line 86 - 'redesaturated' is not a word

The discussion and conclusion are slightly improved.

Reviewer 2 Report

This is a second-reviewed manuscript by this reviewer. The authors argued for all my observations, likewise, they improved the writing of the manuscript based on my observations and perhaps those made by other reviewers. The manuscript improved, the authors present their results and discuss linked to previous investigations carried out by some of them; although it still seems to me that the manuscript might not be clear if the reader is not familiar with these previous studies, it is clear that the manuscript was considerably improved and could be useful in the context of the investigations carried out by the authors.

Author Response

Thanks for your comments.

Reviewer 3 Report

Dear Sir,

I am glad to see all the changes done by the authors are satisfactory and there is no more questions on my side.  This is for your information to make the decision. Thanks  

Author Response

Thanks for your comments.